# Qualifying Land Use and Land Cover Dynamics and Their Impacts on Ecosystem Service in Central Himalaya Transboundary Landscape Based on Google Earth Engine

**Changjun Gu** [1,2], **Yili Zhang** [1,2,3], **Linshan Liu** [1,*], **Lanhui Li** [4], **Shicheng Li** [5], **Binghua Zhang** [1,2], **Bohao Cui** [1,2] and **Mohan Kumar Rai** [1,2]

1   Key Laboratory of Land Surface Pattern and Simulation,
Institute of Geographic Sciences and Natural Resources Research, CAS, Beijing 100101, China;
gucj.18b@igsnrr.ac.cn (C.G.); zhangyl@igsnrr.ac.cn (Y.Z.); zhangbh.17b@igsnrr.ac.cn (B.Z.);
cuibh.19b@igsnrr.ac.cn (B.C.); mkrai2019@igsnrr.ac.cn (M.K.R.)
2   College of Resources and Environment, University of Chinese Academy of Sciences, Beijing 100049, China
3   CAS Center for Excellence in Tibetan Plateau Earth Sciences, Beijing 100101, China
4   Fujian Key Laboratory of Pattern Recognition and Image Understanding, Xiamen University of Technology, Xiamen 361024, China; lilh.15b@igsnrr.ac.cn
5   Department of Land Resource Management, School of Public Administration, China University of Geosciences, Wuhan 430074, China; lisc@cug.edu.cn
*   Correspondence: liuls@igsnrr.ac.cn

**Abstract:** Land use and land cover (LULC) changes are regarded as one of the key drivers of ecosystem services degradation, especially in mountain regions where they may provide various ecosystem services to local livelihoods and surrounding areas. Additionally, ecosystems and habitats extend across political boundaries, causing more difficulties for ecosystem conservation. LULC in the Kailash Sacred Landscape (KSL) has undergone obvious changes over the past four decades; however, the spatiotemporal changes of the LULC across the whole of the KSL are still unclear, as well as the effects of LULC changes on ecosystem service values (ESVs). Thus, in this study we analyzed LULC changes across the whole of the KSL between 2000 and 2015 using Google Earth Engine (GEE) and quantified their impacts on ESVs. The greatest loss in LULC was found in forest cover, which decreased from 5443.20 km$^2$ in 2000 to 5003.37 km$^2$ in 2015 and which mainly occurred in KSL-Nepal. Meanwhile, the largest growth was observed in grassland (increased by 548.46 km$^2$), followed by cropland (increased by 346.90 km$^2$), both of which mainly occurred in KSL-Nepal. Further analysis showed that the expansions of cropland were the major drivers of the forest cover change in the KSL. Furthermore, the conversion of cropland to shrub land indicated that farmland abandonment existed in the KSL during the study period. The observed forest degradation directly influenced the ESV changes in the KSL. The total ESVs in the KSL decreased from 36.53 × 10$^8$ USD y$^{-1}$ in 2000 to 35.35 × 10$^8$ USD y$^{-1}$ in 2015. Meanwhile, the ESVs of the forestry areas decreased by 1.34 × 10$^8$ USD y$^{-1}$. This shows that the decrease of ESVs in forestry was the primary cause to the loss of total ESVs and also of the high elasticity. Our findings show that even small changes to the LULC, especially in forestry areas, are noteworthy as they could induce a strong ESV response.

**Keywords:** land use and land cover; ecosystem service value; Google Earth Engine (GEE); forest fragmentation; transboundary landscape; Himalaya





## 1. Introduction

Ecosystem services can be defined as the benefits that humans gain from ecological processes that contribute to human well-being [1–4]. However, global ecosystem services have been altered by human activities over the past few centuries [5]. Anthropogenic activities can be found in almost every corner of the globe after the onset of the Anthropocene and have emerged as a global driver rapidly sculpturing the ecosystem [6–8].

According to Costanza et al. [9], 60% of worldwide ecosystem services have degraded over the past several decades. Land use and land cover (LULC) changes, mainly driven by human activities [10], are considered to be one of the greatest and most immediate threats affecting ecosystem services [11,12]. LULC changes have thus been considered an important research topic with regard to global environmental change and sustainable development [13–16]. Mountain ecosystems are rich sources of biodiversity [17] and host high plant endemism [18]. They also provide diverse ecosystem services [19]. On the other hand, mountain regions are fragile areas that are sensitive to external forces [20]. Human-driven LULC changes are considered to be among the greatest ecological pressures in mountain regions [21].

As a typical mountain system, the Hindu Kush Himalayan (HKH) region extends ca. four million square kilometers, encompassing eight countries: Afghanistan, Bangladesh, Bhutan (all), China, India, Myanmar, Nepal (all), and Pakistan [22,23]. It is the source of ten major river systems, which provide water, ecosystem services, and the basis for livelihoods to a population of around 210.53 million people in the region [24]. Harboring four of 36 global biodiversity hotspots [25], it provides habitats for numerous wild species but is deeply threatened. The region is extremely fragile in terms of land cover diversity and its association with variable terrain, climate, and sociodemographic interactions. The HKH region is significantly rich in terms of biodiversity but is also one of the least studied in the world [26,27]. The fourth and fifth reports of the Intergovernmental Panel on Climate Change (IPCC) explicitly pointed to the HKH as a data deficient area [28,29].

Even though 39% of the HKH region's land is divided into protected areas to support better conservation [30], the effectiveness of protected areas still faces challenges [31,32]. Almost one-third of the protected areas are transboundary and in these areas, as elsewhere in the HKH region, ecosystems and habitats extend across political boundaries [23]. When conservation policies meet with the administrative and political borders in the territory, the situation becomes more complex because of the nonconformity between natural ecological boundaries and administrative borders [33]. This means that landscape-level planning is necessary and management requires regional cooperation if the ecosystems or habitats are transboundary in nature [34]. For better conservation, seven transboundary landscapes have been identified across the HKH region—based on biodiversity significance, representation of ecoregions, cultural significance, and contiguity of ecosystems for conservation and sustainable development of the region [35]—and are being used to develop transboundary landscape-level planning and management approaches.

The Kailash Sacred Landscape (KSL) is one of the seven transboundary landscapes, named after the Mount Kailash, which is seen as the holiest shrine for several religions [36]. Three of Asia's great rivers have their sources in the landscape: the Indus, the Brahmaputra, and the Ganges River, which provide essential transboundary ecosystem goods and services, both locally and downstream [37]. However, increasingly frequent human activities, together with climate change, have caused rapid land use and land cover changes over the past decades. Uddin et al. [23] have shown the forest fragmentation in Nepal's Kailash Sacred Landscape from 1990 to 2009 and further predicted the future LULC in 2030. Duan et al. [38] assessed LULC changes in the Kailash Sacred Landscape of China from 1990–2008 and quantified driving forces. Singh et al. [39] studied the LULC changes in the Kailash Sacred Landscape of China from 1976–2011 and also pointed out forest fragmentation in the Indian part. All of these studies assessed the LULC changes in three countries using different data sources, study periods, classification systems, and methods. It is almost impossible to compare the differences in LULC changes among the three countries. In short, LULC data covering the entire area are still unclear.

A detailed and accurate knowledge of land cover is crucial for many scientific and operational applications and, as such, it has been identified as an Essential Climate Variable [40]. The development of remote sensing provided an important tool to explore historical and current land cover information at the local, national, regional, and global levels [41]. The complicated process of processing satellite imagery and the high cost of

computing power has limited the relevant research. Google Earth Engine (GEE) provides a high-performance cloud-based platform and access for any researcher [42,43]. GEE houses a massive imagery data collection, including Landsat, MODIS (Moderate Resolution Imaging Spectroradiometer), and Sentinel that can be directly accessed using the JavaScript code within minutes, allowing users to interactively test and develop algorithms and preview results in real time without downloading any images [44]. Furthermore, GEE offers a packaged algorithm for image preprocessing and machine learning classifiers. The efficiency of GEE has been demonstrated by recent studies, including with regard to vegetation change detection [45,46], urban area mapping [47,48], agricultural land mapping [49], grassland monitoring [50,51], extraction of water bodies [52,53], and disaster monitoring [54].

Hence, we used satellite images and GEE to assess LULC changes and examine their impacts on ecosystem service values (ESVs) in the KSL between 2000 and 2015. Our main objectives were to explore: (1) the dynamics of LULC between 2000 and 2015; (2) the ESV changes based on LULC; and (3) their implications for landscape conservation and sustainable land use. This study is expected to provide insights into sacred landscape conservation for future land management.

## 2. Materials and Methods

### 2.1. Study Area

The Kailash Sacred Landscape is located between 79°40′ E–82°30′ E and 29°10′ N–31°20′ N (Figure 1). Mount Kailash, which is considered by multiple religions as the center of the universe, and Lake Manasarovar are the most prominent features in the KSL. There are two sacred lakes near Mount Kailash, Lake Manasarovar and Lake Rakshastal. The region covers an area of over 31,000 km$^2$, including parts of far-western Nepal, northern India, and Purang County, Tibet Autonomous Region of China [23,38,39]. The elevation drop from the highest mountain, Naimona'nyi, to the southern parts is over 7000 m. This loss in elevation causes abundant vegetation types, ranging from tropical broadleaved forest to alpine steppe. Diverse ecosystems provide habitats with rich biodiversity. The landscape is also home to 93 mammal species, 497 bird species, and 134 fish species, among other fauna, making it one of the ecologically richest areas in the western Himalayas [37].

Over a million people live within the landscape and most of this population resides in India and Nepal, with very few persons inhabiting the sparsely populated high-elevation areas on the Tibetan Plateau [37]. Local people rely heavily on the natural resources of this region. In KSL-China, grazing is the primary mode of utilization of grassland, often exerting pressures on fragile ecosystems. Agriculture accounts for a relatively small proportion of land use. In KSL-Nepal and KSL-India, forests cover large parts of these two regions and offer livelihoods to the local people while simultaneously supporting biodiversity conservation. Deforestation and fragmentation because of cropland expansion, infrastructure construction, and illegal timber harvesting have been reported in these regions. Forest cover loss and fragmentation are regarded as main causes of global ecosystem degradation [56]. Accordingly, human activities pose a serious threat to the fragile ecosystems in the KSL.

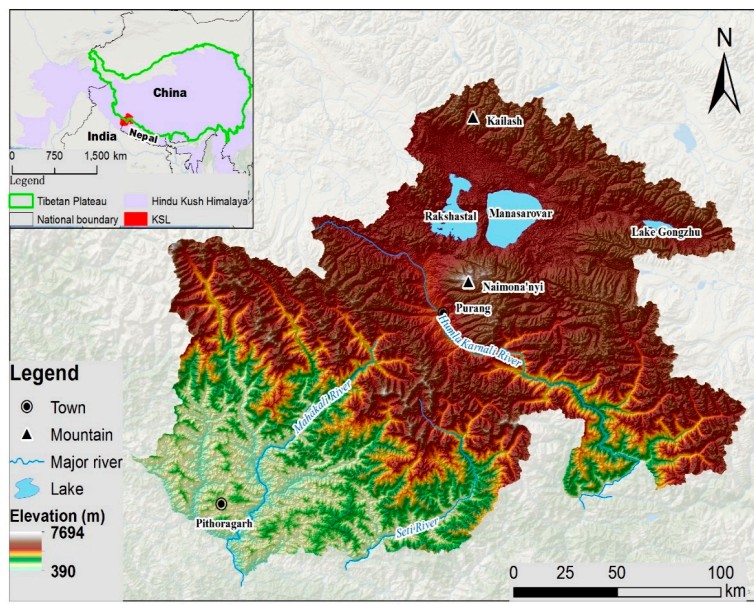

**Figure 1.** Map showing the location and topographic features of Kailash Sacred Landscape (KSL). The Hindu Kush Himalaya (HKH) boundary was obtained from https://rds.icimod.org/home/datadetail?metadataid=3924 (accessed on 8 February 2021) and the Tibetan Plateau boundary from Zhang et al. 2014 [55].

## 2.2. Classification System and Training Data Collection

Land cover classification systems have been defined separately in KSL-China, KSL-Nepal, and KSL-India. Duan et al. [38] classified land cover in KSL-China into ten types: barren land, cropland, desert, glacier, wetland, water bodies, built-up land, low coverage rangeland, medium coverage rangeland, and high coverage rangeland. Uddin et al. [23] divided the land cover in the KSL-Nepal into seven types: forest, shrub land, grassland, cropland, barren area, water bodies, and snow/glacier. Singh et al. [39] classified the land cover system for KSL-India into seven types: forest, settlement, water, agriculture, grassland, scrubland, and snow. The landscape in KSL-China differs from that in KSL-Nepal and KSL-India, thus resulting in different land cover systems. Even though there is a certain resemblance in landscape between KSL-Nepal and KSL-India, differences exist in the classification systems. Following previous frameworks [57–59], we defined our land cover classification system as shown in Table 1. Land cover classes were defined through visual interpretation of high-resolution imagery available from Google Earth, using Landsat images as a reference. Visual interpretation of reference imagery was based on elements that help identify land cover features such as location, size, shape, tone/color, shadow, texture, and pattern [60]. Furthermore, considering the time intervals defined in this study, the training points that were stable during the study period were selected. Finally, we obtained all the defined land cover types and training points shown in Table 1 and Figure S1.

**Table 1.** Classification system and list of training points.

| Land Cover Code | Land Cover Type | Number of the Training Points |
|:---:|:---:|:---:|
| 1 | Water bodies | 165 |
| 2 | Snow/glacier | 255 |
| 3 | Forest | 182 |
| 4 | Built-up area | 80 |
| 5 | Shrub land | 113 |
| 6 | Cropland | 194 |
| 7 | Grassland | 439 |
| 8 | Barren land | 285 |
| 9 | Wetland | 89 |

### 2.3. Preprocessing of the Landsat Images

The Landsat-5 Thematic Mapper (TM), Landsat-7 Enhanced Thematic Mapper Plus (ETM+), and Landsat-8 Operational Land Imager (OLI) top-of-atmosphere (TOA) reflectance products were used for land cover change analysis (available online: https://earthengine.google.com (accessed on 8 February 2021)). The Landsat datasets covering our study area were then imported as image collections into GEE, a cloud-based geospatial analysis platform, for subsequent preprocessing tasks. Preprocessing methods presented by Alban et al. (2018) [61] were modified and applied in this study. The main preprocessing functions, including cloud masking, shadow masking, adding spectral index, etc., were packaged together. Using pixel-based image compositing methods, the best available observations from multiple Landsat images were selected to generate high-quality Landsat image composites for 2000 and 2015 [62–65]. Users can define parameters according to their own requirements, including location of the study area, composite years, cloud cover threshold, etc. The detailed parameters used in this study can be found in the supplementary materials. The quality of the image always suffers from high cloud cover, resulting in empty pixels or scenes. To solve this problem, we combined two strategies. First, we set the combine year parameter to three years to obtain as many images as possible; then we applied the focal_mean function offered by GEE (available at: https://developers.google.com/earth-engine (accessed on 8 February 2021)), a morphological mean filter, to each band of an image using a custom kernel (Figure S2). The detailed parameters used in this study can be found here: https://code.earthengine.google.com/17f98d1e3fe5b7c0e6b432480a65dc9b (accessed on 8 February 2021).

### 2.4. Classification Features Input and Classifier

Multiple spectral indices have been developed to establish the relationship between the spectral and radiometric responses measured by remote sensors and the presence of various land covers, especially vegetation [66]. Huang et al. [67] used the B2–B7 and nominalized difference vegetation index (NDVI) bands as predicting bands for mapping land cover changes in Beijing; Teluguntla et al. [68] used the B2–B7 and NDVI bands as the classification features to map the 30-m cropland extent in Australia and China; and Xiong et al. [69] used B2–B4, B8, and NDVI bands as the predicting bands to acquire a 30-m resolution cropland extent map of continental Africa. Tsai et al. [44] mapped the LULC in Fanjingshan National Nature Reserve using the Landsat spectral band together with the NDVI, normalized difference blue and red (NDBR), normalized difference green and red (NDGR), normalized difference shortwave infrared and near-infrared (NDII), modified soil-adjusted vegetation index (MSAVI), and spectral variability vegetation index (SVVI).

To obtain the most suitable predicting bands, we added spectral bands as below: the B2–B7 and temp bands were selected as the main classification feature inputs, together with 15 spectral indices derived from the Landsat data, including the NDVI [70], the land surface water index (LSWI) [71], the nominalized difference snow index (NDSI) [72], the enhanced vegetation index (EVI) [73], the normalized difference tillage index (NDTI) [74], the normalized difference moisture index (NDMI) [75], the normalized burn ratio (NBR) [76], the vegetation index green (VIG) [77], tasseled cap transformation [78], and other spectral index (SI). These indices were defined as follows:

$$\text{NDVI} = \frac{\rho NIR - \rho \text{Red}}{\rho NIR + \rho \text{Red}} \tag{1}$$

$$\text{LSWI} = \frac{\rho \text{NIR} - \rho \text{SWIR1}}{\rho \text{NIR} + \rho \text{SWIR1}} \tag{2}$$

$$\text{NDSI} = \frac{\rho \text{Green} - \rho \text{SWIR1}}{\rho \text{Green} + \rho \text{SWIR1}} \tag{3}$$

$$\text{EVI} = \frac{\rho NIR - \rho\text{Red}}{\rho NIR + 6 \times \rho\text{Red} - 7.5 \times \rho\text{Blue} + 1} \tag{4}$$

$$\text{NDTI} = \frac{\rho\text{SWIR1} - \rho\text{SWIR2}}{\rho\text{SWIR1} + \rho\text{SWIR2}} \tag{5}$$

$$\text{NDMI} = \frac{\rho NIR - \rho\text{SWIR1}}{\rho NIR + \rho\text{SWIR1}} \tag{6}$$

$$\text{NBR} = \frac{\rho NIR - \rho\text{SWIR2}}{\rho NIR + \rho\text{SWIR2}} \tag{7}$$

$$\text{VIG} = \frac{\rho\text{Green} - \rho\text{Red}}{\rho\text{Green} + \rho\text{Red}} \tag{8}$$

$$\text{SI} = \frac{\rho\text{Red} - \rho\text{Blue}}{\rho\text{Red} + \rho\text{Blue}} \tag{9}$$

where $\rho NIR$, $\rho$Red, $\rho$Green, $\rho$SWIR1, $\rho$SWIR2, and $\rho Blue$ represent the surface reflectance values of the near-infrared band (0.76–0.9 μm), the red band (0.63–0.69 μm), the green band (0.52–0.6 μm), the shortwave infrared band 1 (1.55–1.750 μm), the shortwave infrared band 2 (2.11–2.29 μm), and the blue band (0.45–0.52 μm) for a given pixel, respectively. Furthermore, we also took the topographical factors (slope, elevation, and aspects, available at: https://developers.google.com/earth-engine/datasets/catalog (available in EE as USGS/SRTMGL1_003) (accessed on 8 February 2021)) and nighttime data (available at: https://developers.google.com/earth-engine/datasets/catlog/NOAA_VIIRS_DNB_MONTHLY_V1_VCMSLCFG (accessed on 8 February 2021)) into consideration to better depict cropland and urban areas. We obtained a total of 24 features.

GEE provides 21 classifiers of which random forest (RF) is one of the most widely used as it yields higher classification accuracies, requires less model training time, and is less sensitive to training sample qualities compared to support vector machine (SVM) and artificial neural network (ANN) classifiers [79,80]. In this study, the RF classifier in GEE was trained using 70% of the training data randomly selected and extracted from the sets of image stacks, with the remaining 30% of the training data used for the model validation. A confusion matrix was implemented to assess the accuracy of the classified image with the independent set of ground truth points [81]. The overall accuracy was calculated in GEE, together with the producer's accuracy (PA) and user's accuracy (UA) of each land cover type. A previous study indicated that that the accuracy of a LULC map should higher than 85% for optimal interpretation and identification [82]. To deal with salt and pepper noise, classified images were postprocessed with a majority filter to smooth isolated pixels [83,84]. The overall levels of accuracy for 2000 and 2015 were 88.6% and 89.42%, respectively. The RF classifier produced overall acceptable levels of accuracy for the four classification points in time and the defined LULC types (Tables S1 and S2).

### 2.5. Detection of LULC Changes and Estimation of ESVs

The LULC changes can be calculated using Equation (10). To identify the main conversion directions and highlight the dominant dynamic events in land use/cover changes, we used ArcGIS (version 10.4) to generate the transfer matrix for each period and visualized the transfer process with a Sankey Diagram (available online at: https://sankey.csaladen.es (accessed on 8 February 2021)) [85,86]. The calculation is as follows:

$$\text{R} = \frac{L_t - L_{t-1}}{L_{t-1} * \Delta t} \times 100\% \tag{10}$$

where R represents the LULC change rate, $L_t$ represents land cover type in year $t$, $L_{t-1}$ represent land cover in the most recent time interval, and $\Delta t$ denotes the time interval (15 in this study).

To better understand the consequences of the conversion from forest to other LULC types, we further assessed the forest fragmentation of the KSL in 2000 and 2015 following

the method described by Vogt et al. [87]. The forest LULC type was divided into six classes, patch, edge, perforated, small core (SC) (<250 acres), medium core (MC) (250–500 acres), and large core (LC) (>500 acres), by computing the distance from forest pixels to non-forest pixels. We defined the edge width as 100 m in reference to previous studies [23].

Costanza et al. [1] presented a model to estimate global ecosystem service value. However, this estimation method is best suited for Western countries; Xie et al. [88] therefore grouped the ESVs into four types and nine subtypes specific to China on this basis and using data from [5]. Costanza et al. [9] further presented a new method for the estimation of global ESVs and found that the ESVs of certain land cover types increased (e.g., the ESVs of forest land cover increased by 2462 USD per hectare per year from 1997–2011) while the remaining land cover types remained stable. In this study, we adopted the same equivalent value as that used by Song et al. [89] (Table 2). The equations used to evaluate the KSL's ESVs and their changes are as follow:

$$\text{ESV}_t = \sum_{i=1}^{n} Area_i \times ESV_i \tag{11}$$

$$C_{\Delta t} = \frac{E_{end} - E_{start}}{E_{Start}} \times 100\% \tag{12}$$

where $\text{ESV}_t$ denotes the total ESV at time t (2000, 2005, 2010, 2015); $Area_i$ represents the area of land cover i, $ESV_i$ represents the ESV of land cover I, and n denotes the total number of land cover types (seven types after reclassification in this study). $C_t$ represents the changes in ESV within a time interval (e.g., 2000–2005) and $E_{end}$ and $E_{start}$ denote the ESVs at the end and start of the time interval, respectively.

**Table 2.** Ecosystem service values (ESVs) of land cover types defined in this study.

| Land Cover Defined in This Study | Equivalent Biome (Song et al. 2017) [89] | ESVs Per Unit Area ($/hm$^2$/year) |
|---|---|---|
| Water bodies Snow/glacier | Water areas | 2607.77 |
| Forest Shrub land | Forestry areas | 1616.99 |
| Grassland | Grassland | 671.06 |
| Cropland | Cultivated land | 454.28 |
| Built-up area | Built-up areas | 0 |
| Barren land | Unused land | 79.93 |
| Wetland | Wetland | 3149.45 |

*2.6. Elasticity of ESV Changes in Response to LULC Changes*

For the purpose of investigating the relation between LULC and ESVs, elasticity as defined by Song et al. [89] was applied in this study. The concept of elasticity is used to measure the sensitivity of a variable to change in another variable. Here, elasticity was used to measure the percentage change in ESV in relation to the percentage change in LULC, and thus can be described as follows:

$$\text{EEl} = \left| \frac{(E_{\text{end}} - E_{\text{start}})/E_{\text{start}} \times 100\%}{LCP} \right| \tag{13}$$

$$LCP = \frac{\sum_{i=1}^{7} \Delta LUT_i}{\sum_{i=1}^{7} LUT_i} \tag{14}$$

where EEI represents the elasticity of ESV change in response to changes in LULC, $E_{end}$ is the ESV at the end of the study period, $E_{start}$ is the ESV at the beginning of the study period, LCP is the conversion percentage of land (which reveals both speed and degree of

LULC conversion), $\Delta LUT_i$ is the converted area of the *i* type of LULC, $LUT_i$ is the area of the *i* type of LULC, and *T* is the time gap (in years) of the study period.

## 3. Results

### 3.1. The Spatial Distribution of LULC and Its Changes

As shown in Figure 2, there was a significant difference in land cover between the Himalayan northern slopes (China side) and southern slopes (Nepal and India side). Statistical results indicated that most land in the KSL was covered by grassland (23.98% in 2000, 25.74% in 2015) followed by barren land (21.34% in 2000, 21.98% in 2015), and forest (17.45% in 2000, 16.04% in 2015) (Table 3). Grassland was mainly distributed in KSL-China and widely distributed on the Tibetan Plateau (55.73% in 2000, 52.84% in 2015). Over 60% of barren land was distributed in KSL-China and forest land cover was mainly distributed in KSL-Nepal and KSL-India (60.69% and 39.31% in 2000, respectively). Snow/glacier accounted for more than 15.16% of the total area and over 53% of snow/glacier was distributed in KSL-Nepal (54.14% in 2000, 53.61% in 2015). Cropland and built-up areas were the main land cover types closely relevant to human activities and these were mainly distributed in KSL-Nepal and KSL-India.

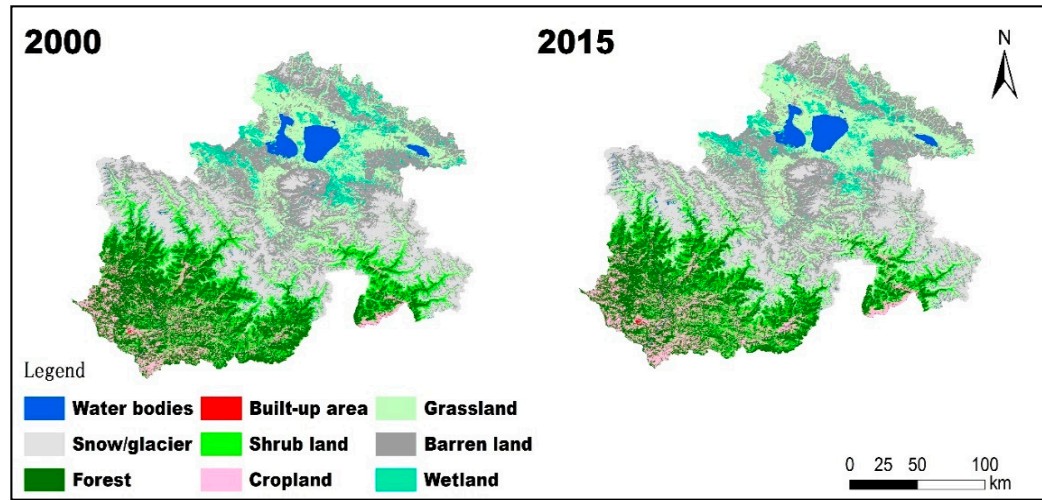

**Figure 2.** The distribution of different land cover types in KSL in 2000 and 2015.

**Table 3.** Land use and land cover (LULC) in the KSL during the period 2000–2015.

| Land Cover | Area in 2000 (km$^2$) | % | Area in 2015 (km$^2$) | % | Changed Area (2000–2015) | Change Rate (2000–2015) |
|---|---|---|---|---|---|---|
| Water bodies | 990.27 | 3.17 | 994.71 | 3.19 | 4.43 | 0.03 |
| Snow/glacier | 4728.51 | 15.16 | 4687.55 | 15.03 | −40.96 | −0.06 |
| Forest | 5443.20 | 17.45 | 5003.37 | 16.04 | −439.82 | −0.54 |
| Built-up area | 65.59 | 0.21 | 66.05 | 0.21 | 0.46 | 0.05 |
| Shrub land | 2917.78 | 9.35 | 2528.17 | 8.11 | −389.61 | −0.89 |
| Cropland | 1910.59 | 6.13 | 2257.50 | 7.24 | 346.90 | 1.21 |
| Grassland | 7479.89 | 23.98 | 8028.35 | 25.74 | 548.46 | 0.49 |
| Barren land | 6655.26 | 21.34 | 6854.46 | 21.98 | 199.20 | 0.20 |
| Wetland | 1000.04 | 3.21 | 770.98 | 2.47 | −229.07 | −1.53 |
| Total | 31,191.13 | 100 | 31,191.13 | 100 | | |

Between 2000 and 2015, four land cover types showed decreasing trends and the other five land cover types showed increasing trends (Table 3). The greatest loss was found for forest: a total of 439.82 km$^2$ forest cover loss was observed in the KSL. The decrease of forest cover in KSL-Nepal contributed 89.68% of the total forest loss during the research period. Shrub land also showed an obvious decreasing trend, with a total loss of 389.61 km$^2$ during

the research period, decreasing at a rate of 0.89% per year. The decrease of shrub land in KSL-Nepal and KSL-India contributed 65.55% and 34.3%, respectively, to the total shrub land loss. During the research period, wetland and snow/glacier decreased by 229.07 km$^2$ and 40.96 km$^2$, respectively. Among the land cover types with increasing trends, the biggest gains were found in grassland: grassland increased by 548.46 km$^2$ during the research period. The increase in grassland in KSL-Nepal and KSL-India contributed 58.32% and 28.22% to the total gains in grassland in the KSL, respectively. From 2000–2015, cropland increased by 346.90 km$^2$ and at a rate of 1.21% per year. The biggest increase was observed in KSL-Nepal, where cropland increased by 247.94km$^2$ from 2000–2015. Barren land was found to increase from 6655.26 km$^2$ to 6854.46 km$^2$ between 2000 and 2015 and at a rate of 0.2% per year. Changes in water bodies and built-up areas were not obvious and only increased by 4.43 km$^2$ and 0.46km$^2$, respectively, during the research period. The results indicate that the area of land types with higher ecosystem service values (e.g., forest, shrub land, and wetland) decreased.

Forest, barren land and grassland were significantly converted to other land cover types in the period from 2000–2015 (Table 4 and Figure 3). A total of 857.81 km$^2$ of forest were converted to other land cover types, including 59.18% that were converted to shrub land and 34.19% that were converted to cropland. This indicates the forest fragmentation occurred between 2000 and 2015. About 1125.07 km$^2$ of barren land were converted into other land cover types during the research period, 39.96% of which were converted to snow/glacier. Snow/glacier mainly converted to barren land during the research period: a total of 526.78 km$^2$ of snow/glacier were converted to barren land. About 1150.91 km$^2$ of grassland were converted to other land cover types with 61.17% converted to barren land. During the study period, shrub land contributed most to the expansion of cropland: a total of 425.48 km$^2$ of shrub land was converted to cropland. Meanwhile, cropland mainly converted to shrub land and forest between 2000 and 2015: a total of 288.38 km$^2$ and 170.74 km$^2$ of cropland converted to shrub land and forest, respectively. The expansion of built-up areas was mainly at the cost of cropland. The results indicate that deforestation and cropland abandonment occurred in KSL-Nepal and KSL-India simultaneously.

**Table 4.** Transition matrix of different LULC types in the KSL during the period 2000–2015.

| 2015 | 2000 | | | | | | | | |
|---|---|---|---|---|---|---|---|---|---|
| | Water Bodies | Snow/ Glacier | Forest | Built-up Area | Shrub Land | Cropland | Grassland | Barren Land | Wetland |
| Water bodies | 832.23 | 61.02 | 19.24 | 4.54 | 4.40 | 0.91 | 32.87 | 35.5 | 0.00 |
| Snow/glacier | 48.92 | 4068.60 | 2.66 | 2.43 | 3.55 | 0.27 | 76.05 | 526.78 | 0.70 |
| Forest | 7.87 | 0.09 | 4586.69 | 8.21 | 507.66 | 293.25 | 39.95 | 0.79 | 0.00 |
| Built-up area | 7.74 | 0.06 | 6.49 | 23.11 | 1.40 | 25.02 | 1.39 | 0.49 | 0.00 |
| Shrub land | 16.12 | 31.59 | 203.00 | 0.85 | 1641.12 | 425.48 | 588.14 | 11.69 | 0.00 |
| Cropland | 1.59 | 0.02 | 170.74 | 10.41 | 288.38 | 1420.75 | 17.31 | 1.62 | 0.24 |
| Grassland | 29.40 | 77.80 | 13.99 | 8.32 | 79.91 | 75.55 | 6329.91 | 704.02 | 161.93 |
| Barren land | 51.32 | 449.60 | 1.83 | 8.22 | 1.94 | 3.48 | 555.73 | 5531.32 | 52.95 |
| Wetland | 0.00 | 0.32 | 0.00 | 0.03 | 0.00 | 13.31 | 387.99 | 43.23 | 555.29 |

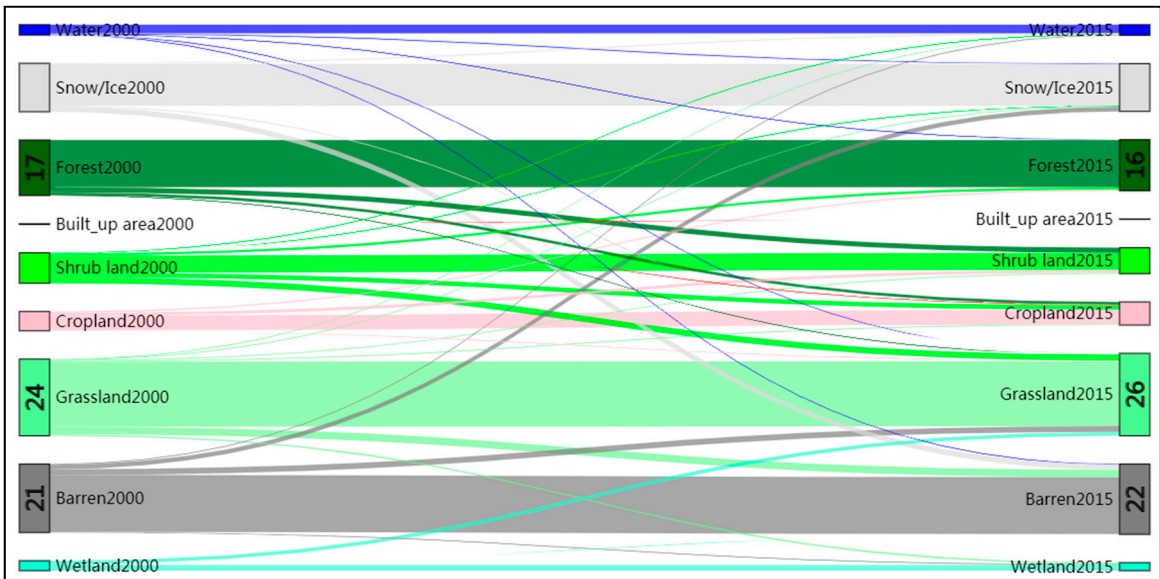

**Figure 3.** Sankey diagram of LULC transitions in the KSL during the period 2000–2015. The map was created using the tool at https://sankey.csaladen.es/#.

### 3.2. Forest Fragmentation in the KSL

A conversion of forest to other land cover types indicating forest fragmentation occurred in the KSL during the research period (Table 5). The distribution of and changes in forest fragmentation in 2000 and 2015 are depicted in Figure 4. During the study period, the forest fragmentation changed significantly. In 2000, core forest (>500 acres) covered 34.24% of the forest area, followed by edge forest covering 30.39%, perforated forest covering 18.44%, core forest (<250 acres) covering 7.72%, patch forest covering 5.95%, and core forest (250–500 acres) covering 2.89%. In 2015, edge forest covered 33.67% of the forest area, followed by core forest (>500 acres) covering 28.1%, perforated forest covering 17.22%, patch forest covering 8.69%, core forest (<250 acres) covering 8.53%, and core forest (250–500 acres) covering 3.79%. Core forest (>500 acres) decreased from 1883.90 km$^2$ to 1406.05 km$^2$, with a change rate of 25.36%. Meanwhile, patch forest increased from 323.81 km$^2$ to 434.83 km$^2$, with a change rate of 34.29%.

**Table 5.** Forest fragmentation and change in KSL between 2000 and 2015.

| Type of Patches | 2000 (km$^2$) | 2015 (km$^2$) | 2000–2015 (km$^2$) | Change Rate (%) |
|---|---|---|---|---|
| Patch | 323.81 | 434.83 | 111.02 | 34.29 |
| Edge | 1654.24 | 1684.39 | 30.15 | 1.82 |
| Perforated | 1003.80 | 861.59 | −142.21 | −14.17 |
| Core (<250 acres) | 420.12 | 426.81 | 6.69 | 1.59 |
| Core (250–500 acres) | 157.33 | 189.71 | 32.37 | 20.58 |
| Core (>500 acres) | 1883.90 | 1406.05 | −477.85 | −25.36 |

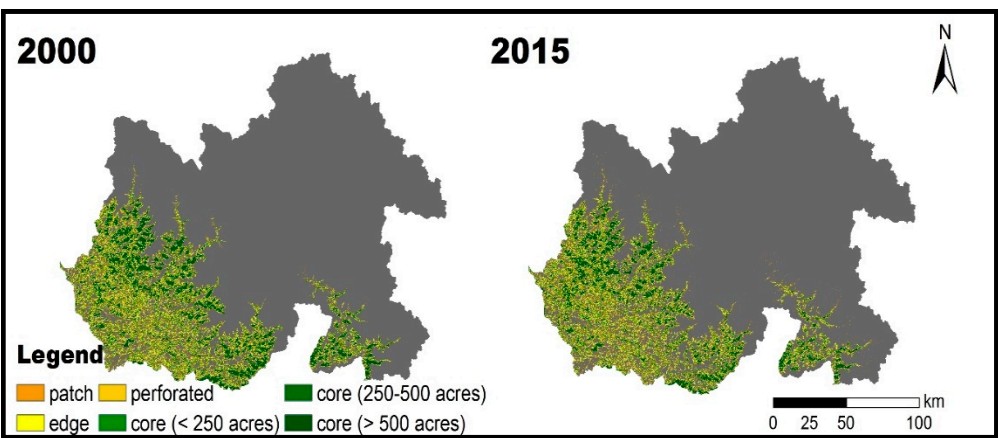

**Figure 4.** Forest fragmentation maps from 2000 and 2015. Core—relatively distant from the forest–non-forest boundary; patch—forests too small to be considered as core forest; perforated—boundaries between core forest and relatively small perforations; edge—boundaries of relatively large perforations and the exterior boundaries of core forest regions.

Forest land cover was mainly distributed in KSL-Nepal; this also true for the core forest (>500 acres), covering 57.68% of the total core forest (>500 acres) in KSL. In 2000, core forest (>500 acres) covered 32.90% of the forest area in KSL-Nepal, followed by edge forest covering 31.23%, perforated forest covering 18.28%, core forest (<250 acres) covering 8.24%, patch forest covering 6.12%, and core forest (250–500 acres) covering 3.23%. During the research period, core forest (>500 acres) decreased by 31.97% and patch forest increased by 38.62%. The rest of the forest land cover was distributed in KSL-India. In 2000, core forest (>500 acres) covered 37.25% of the forest area in KSL-India, followed by edge forest covering 29.09%, perforated forest covering 18.70%, core forest (<250 acres) covering 6.91%, patch forest covering 5.68%, and core forest (250–500 acres) covering 2.36%. From 2000–2015, core forest (>500 acres) decreased by 16.36% and core forest (250–500acrs) increased by 56.59%. The results suggest that deforestation and forest fragmentation occurred in KSL, especially in KSL-Nepal, during the research period.

### 3.3. The LULC Changes in KSL-China, KSL-Nepal, and KSL-India

In 2015, most land in the three countries was covered by different land cover types (Table 6). In KSL-China, barren land accounted for the largest proportion of land cover. During the research period, no evident changes were observed in barren land (increase of 2.43 km$^2$). As the second largest land cover type, grassland increased from 4168.41 km$^2$ to 4242.24 km$^2$ during the research period. Between 2000 and 2015, snow/glacier increased from 756.58 km$^2$ to 815.46 km$^2$, an increasing trend opposite to the broader picture for KSL snow/glacier. A great increase was observed in cropland, which increased by 73.07 km$^2$ between 2000 and 2015. Wetland, water bodies and shrub land showed decreasing trends during the research period, while the largest decrease was found in wetland (decreased by 203.29 km$^2$). In KSL-Nepal, grassland, cropland, and barren land contributed most to land cover increases. During the research period, the greatest gains were found in grassland, which increased by 319.87km$^2$, followed by cropland, which contributed the most to the cropland expansion in the KSL (increase of 247.94 km$^2$, accounting for over 70% of the total increase). Forest in KSL-Nepal decreased from 3303.37 km$^2$ to 2908.90 km$^2$ during the research period. Between 2000 and 2015, shrub land decreased by 255.43 km$^2$, second only to the loss of forest. Snow/glacier showed a decreasing trend and decreased by 47.31 km$^2$ during the research period. In KSL-India, the greatest gains were found for grassland, which increased by 154.76 km$^2$ between 2000 and 2015. Changes in cropland were not evident, with an increase from 796.86 km$^2$ to 822.83 km$^2$. Shrub land decreased from 905.79 km$^2$ to 772.13 km$^2$ during the research period. The most significant changes

were observed for cropland expansion and forest loss, which were mainly distributed in KSL-Nepal.

**Table 6.** Dynamic changes in LULC types between 2000–2015.

| Land Cover | KSL-China | | | | KSL-Nepal | | | | KSL-India | | | |
|---|---|---|---|---|---|---|---|---|---|---|---|---|
| | 2000 (km²) | 2015 (km²) | Change Area (km²) | Change Rate (%) | 2000 (km²) | 2015 (km²) | Change Area (km²) | Change Rate (%) | 2000 (km²) | 2015 (km²) | Change Area (km²) | Change Rate (%) |
| Water bodies | 754.55 | 748.52 | −6.03 | −0.05 | 136.44 | 150.07 | 13.64 | 0.67 | 99.45 | 96.30 | −3.15 | −0.21 |
| Snow/glacier | 756.58 | 815.46 | 58.89 | 0.52 | 2560.67 | 2513.36 | −47.31 | −0.12 | 1412.07 | 1359.45 | −52.63 | −0.25 |
| Forest | 0.00 | 0.00 | 0.00 | 0.00 | 3303.37 | 2908.90 | −394.47 | −0.80 | 2140.04 | 2094.67 | −45.38 | −0.14 |
| Built-up area | 0.15 | 1.83 | 1.69 | 75.37 | 32.92 | 32.26 | −0.67 | −0.14 | 32.60 | 32.04 | −0.56 | −0.11 |
| Shrub land | 0.63 | 0.05 | −0.58 | −6.17 | 2011.52 | 1756.09 | −255.43 | −0.85 | 905.79 | 772.13 | −133.66 | −0.98 |
| Cropland | 13.61 | 86.67 | 73.07 | 35.80 | 1100.41 | 1348.35 | 247.94 | 1.50 | 796.86 | 822.83 | 25.97 | 0.22 |
| Grassland | 4168.41 | 4242.24 | 73.83 | 0.12 | 2358.69 | 2678.55 | 319.87 | 0.90 | 952.65 | 1107.41 | 154.76 | 1.08 |
| Barren land | 4227.39 | 4229.81 | 2.43 | 0.00 | 1659.08 | 1801.75 | 142.67 | 0.57 | 767.51 | 821.74 | 54.23 | 0.47 |
| Wetland | 898.35 | 695.06 | −203.29 | −1.51 | 97.66 | 71.42 | −26.24 | −1.79 | 4.08 | 4.49 | 0.41 | 0.68 |

*3.4. The Spatial Distribution of ESVs and Their Response to LULC Changes*

The ESVs of the KSL in 2000 and 2015 were estimated (Figure 5 and Table 7). The total ESV of the KSL in 2000 was $36.53 \times 10^8$ USD $y^{-1}$. During the research period, the total ESV decreased by $1.17 \times 10^8$ USD $y^{-1}$. In general, water areas and forestry areas contributed most to the total ESV, accounting for about 77.83% in 2000 and 76.38% in 2015. In 2000, water areas contributed about 40.82% of the total ESV in the KSL and 41.91% in 2015. Forestry areas contributed the second most to the total ESV and decreased from $13.52 \times 10^8$ USD $y^{-1}$ in 2000 to $12.18 \times 10^8$ USD $y^{-1}$ in 2015. The ESVs of grassland, cultivated land, and unused land showed an increasing trend. The greatest gains were found in grassland: the ESV of grassland increased from $5.42 \times 10^8$ USD $y^{-1}$ to $5.67 \times 10^8$ USD $y^{-1}$ during the research period. With the expansion of cropland, the ESV of cropland increased by $0.16 \times 10^8$ USD $y^{-1}$ from 2000–2015. Although wetland accounted for a small area in the KSL, the high ESV of wetland enlarged its influence on the total ESV. During the research period, the ESV of wetland decreased by $0.16 \times 10^8$ USD $y-^1$, which offset the increase of ESV of cropland. Since the ESV of built-up areas was zero, this kind of land cover contributed no ESV.

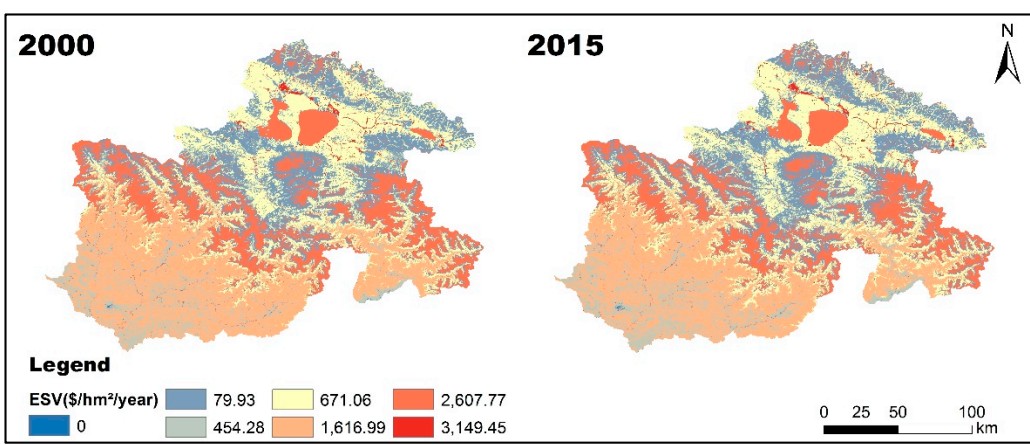

**Figure 5.** The distribution of ESVs in the KSL in 2000 and 2015.

**Table 7.** The ESVs of 11 LULC types in 2000 and 2015.

| Land Cover | Value ($10^8$ USD y$^{-1}$) | | Change Value ($10^8$ USD y$^{-1}$) | Change Rate (%) |
| --- | --- | --- | --- | --- |
| | **2000** | **2015** | **2000–2015** | **2000–2015** |
| Water areas | 14.91 | 14.82 | −0.09 | −0.6 |
| Forestry area | 13.52 | 12.18 | −1.34 | −9.91 |
| Grassland | 5.42 | 5.67 | 0.25 | 4.61 |
| Cultivated land | 0.87 | 1.03 | 0.16 | 18.39 |
| Built-up areas | 0 | 0 | 0 | 0 |
| Unused land | 0.53 | 0.55 | 0.02 | 3.77 |
| Wetland | 1.28 | 1.12 | −0.16 | −12.50 |
| Total | 36.53 | 35.35 | −1.17 | −3.20 |

In KSL-China, the total ESV was $8.44 \times 10^8$ USD y$^{-1}$ in 2000, lower than the ESVs in KSL-Nepal ($18.07 \times 10^8$ USD y$^{-1}$) and KSL-India ($10.03 \times 10^8$ USD y$^{-1}$). Water bodies contributed most to the total ESV in China, accounting for almost half of the total ESV in KSL-China, followed by Grassland ($3.20 \times 10^8$ USD y$^{-1}$). From 2000–2015, the total ESV in KSL-China increased by $0.06 \times 10^8$ USD y$^{-1}$. The increase of water bodies by $0.13 \times 10^8$ USD y$^{-1}$ contributed most to the increase of the total ESV in KSL-China. The decrease of the ESV of grassland was the main cause for the decrease of total ESV in the KSL. The ESV of wetland decreased from $0.95 \times 10^8$ USD y$^{-1}$ in 2000 to $0.93 \times 10^8$ USD y$^{-1}$ in 2015. In KSL-Nepal, the total ESV accounted for about 50% of the total ESV in the KSL. During the research period, ESV in KSL-Nepal decreased by $0.88 \times 10^8$ USD y$^{-1}$; the decrease of the ESVs of forestry areas was the main cause of the loss of total ESV in KSL-Nepal. In 2000, the ESVs of forestry areas accounted for 47.54% of the total ESV in KSL-Nepal. However, this number decreased to 43.87% in 2015. From 2000–2015, the ESV of forestry areas decreased by $1.05 \times 10^8$ USD y$^{-1}$. The ESVs of cultivated land and grassland increased by $0.11 \times 10^8$ USD y$^{-1}$ and $0.21 \times 10^8$ USD y$^{-1}$, respectively, and offset a small part of the ESV loss. In KSL-India, the total ESV decreased from $10.03 \times 10^8$ USD y$^{-1}$ in 2000 to $9.67 \times 10^8$ USD y$^{-1}$ in 2015. The ESV of forestry areas contributed most to the total ESV in KSL-India, similar to KSL-Nepal, followed by water areas. The greatest loss was observed in forestry areas: the ESV of forestry areas decreased by $0.29 \times 10^8$ USD y$^{-1}$. From 2000 to 2015, the ESV of water areas decreased by $0.15 \times 10^8$ USD y$^{-1}$, second only to the loss in ESV of forestry areas. The greatest gains in ESV were found for grassland, which increased by 0.12 during the research period. The small changes of cropland in KSL-India made a relevant but small contribution to the changes in ESV in KSL-India.

The elasticity of ESV change with respect to LULC changes during the research period was 2.33, which indicates that a conversion of 1% of land area would result in an average change of 2.33% in the ESV. The elasticity of the ESVs in the three countries was further calculated. The results show that KSL-China had the highest elasticity at 5.27, indicating that a conversion of 1% of land area would result in an average change of 5.27% in the ESV. In KSL-Nepal, the elasticity was 4.34, higher than that of KSL-India (1.57).

## 4. Discussion

### 4.1. LULC Changes across the KSL

Detailed LULC research is of great significance for managing natural resource effectively [23]. In this study, we applied an RF algorithm to classify the LULC in the KSL in 2000 and 2015 using GEE. For a solution to the problem of the low-quality imagery caused by the high cloud cover in this region, we adopted a pixel-based image composite algorithm and filled the blank pixels using the focal_mean function. Furthermore, the spectral index, terrain factors, and nighttime light data were used to improve the accuracy of the classification. The entire process of LULC classification, except where otherwise noted, was accomplished in GEE. The overall accuracies of the LULC classification in 2000 and 2015 were 87.69% and 85.73%, respectively, indicating the good performance of our

methods. Based on the LULC, we further estimated the ESVs of the KSL and qualified the responses of ESVs to LULC changes.

During the research period, the greatest land cover loss was found for forest cover, which decreased by 439.82 km$^2$. Forest area in Nepal and India decreased from 3303.37 km$^2$ to 2908.90 km$^2$ and from 2140.04 km$^2$ to 2094.67 km$^2$, respectively. The same phenomenon of forest cover decrease has also been found in other Himalayan regions [90,91]. The second greatest land cover loss was found for shrub land, which decreased by 389.61 km$^2$ between 2000 and 2015. These decreasing vegetated areas, especially the forest cover loss, may pose a threat to biodiversity conservation and livelihood [92,93]. Meanwhile, grassland and cropland areas significantly increased during the research period, which is consistent with the findings of Uddin et al.'s [23] and Singh et al. [39]. Cropland increased from 1910.59 km$^2$ to 2257.50 km$^2$, with a change rate of 1.21% per year. The three countries showed the same increasing trend. The largest growth was found in Nepal, where cropland increased by 247.94 km$^2$. Results from the transfer matrix show that expansions of cropland were mainly derived from forest and shrub land in KSL-India and KSL-Nepal. It has been previously shown that expansion of cropland is one of the major drivers of deforestation in the Himalayas [94]. Through statistical analysis of the LULC changes along the elevation, we further found that the increase of cropland was mainly distributed between 1000 and 2500 m in the KSL, accounting for 79.63% of the total increase (Figure S3). The largest growth was found at 1500–2000 m, accounting for 35.01%. The decrease of forest was mainly distributed between 1000 and 3500 m, accounting for 99.28% of the total loss. The most passive change of forest cover was between 1500 and 2000 m. An earlier study has shown that, in the final three decades of the 20th century, forest degradation mainly occurred in temperate oak forests at elevations of 1800–2800 m, with some forests also lost at lower elevations [95]. Lowland areas are considered more favorable for supporting human livelihood and thus result in more intense LULC changes [96].

The forest in the KSL is undergoing a process of fragmentation under the drivers of cropland expansion and illegal timber extraction [23,39]. As an important habitat for countless wild species, the decrease in forest cover along with forest fragmentation put wild life in danger. Sarker et al. (2018) assessed the habitat suitability and connectivity of the common leopard (Panthera pardus) in Kailash Sacred Landscape [97]. Their results show that the best forest connectivity for leopards lies between large forest patches situated at the middle elevational range of the landscape, associated with moderate to medium slopes and a high density of rivers and streams. The decrease in core forest cover may threaten the habitat of the common leopard. Increasing human activities (expansion of cropland and built-up areas) [98,99], together with climate change [100], have resulted in rapid changes in the Himalayan ecosystem [101]. Invasive species are another important issue to consider. Research has shown that species, including invasive species, tend to move to higher elevation regions in global warming contexts [102,103].

The conversion of cropland to forest and shrub land indicates that farmland abandonment occurred in the KSL. Between 2000 and 2015, 288.38 km$^2$ of cropland were converted to shrub land. A noticeable increasing trend in farmland abandonment has been reported all around the world, especially in mountain regions [104,105]. According to a previous study, the hill and mountain regions of the Nepal Himalayas are more prone to farmland abandonment because of labor migration [106–108]. Singh et al. (2015) also found that, in KSL-India, continuous migration forced the conversion of high-altitude agricultural lands into grasslands and scrublands [39]. From the perspective of ecosystems service, farmland abandonment itself has positive effects [109]; however, it also poses a threat to the food security of local livelihoods [107].

*4.2. ESV Changes in Response to LULC Changes*

LULC changes are generally accepted as one of the critical drivers of global change [110]. During the studied 15-year period, the total ESV of the KSL decreased from $36.53 \times 10^8$ USD y$^{-1}$ to $35.35 \times 10^8$ USD y$^{-1}$, decreasing at a rate of 0.21%/year. The decrease of forestry areas was

the primary cause for the loss of ESV. The largest ESVs were observed in KSL-Nepal, due to the large forestry areas, whereas KSL-China was responsible for the smallest proportion of the ESV. However, the ESVs in KSL-China showed an inverse trend compared to KSL-Nepal and KSL-India. Between 2000 and 2015, the total ESV of KSL-China increased by $0.06 \times 10^8$ USD $y^{-1}$ thanks to the increase in water areas. On a global scale, the global terrestrial ESV decreased at a rate of 2.06%/year from 1997 to 2011 [9]. Hence, changes in ESVs in the KSL were more modest than those globally. On a national scale, the terrestrial ESVs in China decreased at a rate of 0.03% per year from 1988–2008. This indicates that the decreases in ESVs in KSL were much more rapid. However, there are large gaps in other Himalayan regions. Bhaskar et al. [111] assessed the ESVs in the Transboundary Karnali River Basin (KRB), Central Himalayas, and showed that they increased by $1.59 \times 10^8$ USD $y^{-1}$ from 2000–2017. Increase of shrub/grassland contributed the most to the increase of ESVs in this region, followed by bare land. Raju et al. [112] estimated the ESVs in the Transboundary Gandaki River Basin (GRB), Central Himalayas, indicating that there was a $1.68 \times 10^8$ USD $y^{-1}$ increase in ESVs from 1990–2015 due to the increase of cropland and forest cover. Zhao et al. [113] assessed the LULC changes and ESVs in the Koshi River Basin (KRB) and found that the latter decreased by $2.05 \times 10^8$ USD $y^{-1}$ from 1990–2010 because of the decrease in forest and glacier cover. Even though large knowledge gaps are still present for different regions, the importance of forest land cover is obvious and changes to it directly affect regional ESVs.

With regard to the elasticity in the KSL, a result of 2.33 indicates that that the conversion of 1% of land area would result in average changes of 2.33% in ESVs. The region where changes in ESVs had the highest elasticity in relation to LULC changes was KSL-China, where the high elasticity of ESV change in relation to LULC changes was attributable to the concentrations of unused land, wetland, and water bodies, the LULC types with the highest and the lowest ESVs. In KSL-Nepal, deforestation was the main cause of the high elasticity. Forest cover in KSL-Nepal accounted for the largest proportion of this type of land cover in the KSL and decreased by 394.47 km$^2$ during the studied 15-year period. The elasticity in KSL-India was relatively small, mainly due to the small decrease of forest cover. High elasticity indicates that even small LULC changes would have serious effects on ESVs.

*4.3. Uncertainty and Limitations of This Study*

In this study, we failed to accurately extract the built-up areas in the KSL because of the limited resolution of Landsat images and relevant small buildings in the KSL mountain regions. To resolve this problem, we tried adding nighttime light data to improve accuracy for built-up areas. However, this approach only works in regions with night lights, such as Pithoragarh (Figure S4). Therefore, the changes to built-up areas in KSL-Nepal and KSL-India showed a slightly decreasing trend. Even so, LULC and ESV changes were not strongly affected due to the small proportion of built-up areas and their ESVs of zero. Long time-series LULC change monitoring can reveal more details behind these changes. Given the available images, we only studied the LULC changes from 2000–2015, and thus LULC change fluctuations may have been hidden. In this study, we adopted the benefit/value transfer method presented by Song et al. [89], though many critiques of the benefit/value transfer method remain unanswered. Biophysical models might be more helpful for analyzing complex ecological systems and their impacts.

## 5. Conclusions

In this study, we extended an LULC study to the entire KSL and further assessed the changes in ESVs between 2000 and 2015. During the study period, the KSL experienced significant LULC changes: forest and shrub land decreased by 439.82km$^2$ and 389.61km$^2$, respectively, whereas grassland and cropland increased by 548.46km$^2$ and 346.90km$^2$, respectively. The conversion of forestry areas to cropland was the main cause of cropland expansion. Meanwhile, the conversion of cropland to shrub land indicates that there

was farmland abandonment in the KSL. The decrease of forestry areas may pose a threat to biodiversity and livelihoods there. During the studied 15-year period, the large core (>500 acre) forest type decreased by 25.36% and patch forest increased by 34.29%. Severe forest fragmentation was observed in the KSL, mainly distributed in KSL-Nepal, leading to a decrease in ESVs in the KSL. Between 2000 and 2015, the total ESV in the KSL decreased by $1.17 \times 10^8$ USD $y^{-1}$ and the ESV of forestry areas decreased by $1.34 \times 10^8$ USD $y^{-1}$. The decrease of ESV in forestry areas contributed most to the loss of total ESV and also to the high elasticity. This study revealed that even small LULC changes can cause relevant high ESV changes in the KSL.

**Supplementary Materials:** The following are available online at https://www.mdpi.com/2073-445X/10/2/173/s1, Figure S1. Training points in Google Earth high-resolution image and Landsat5/7/8 false color composite image; Figure S2. Image with empty pixels (left) and processed using focal_mean function (right); Table S1. The confusion matrix in 2000; Table S2. The confusion matrix in 2015; Figure S3. The LULC changes along the elevation in KSL from 2000–2015; Figure S4. Built-up areas of Pithoragarh in 2000 and 2015.

**Author Contributions:** Y.Z. and L.L. (Linshan Liu) designed the study. C.G. and L.L. (Linshan Liu) conceived the research; C.G. studied and wrote the paper and mapped the land cover data and drew the figures; S.L., L.L. (Lanhui Li), B.Z. and B.C. revised the paper and polished the language. Y.Z. revised the paper and contributed to explanation of the results and the discussion. M.K.R. polished the manuscript. All authors have read and agreed to the published version of the manuscript.

**Funding:** This work was financially supported by the Second Tibetan Plateau Scientific Expedition and Research (Grant No. 2019QZKK0603), the Strategic Priority Research Program of the Chinese Academy of Sciences (Grant No. XDA20040201), and the National Natural Science Foundation of China (Grant No. 41761144081).

**Institutional Review Board Statement:** Not applicable.

**Informed Consent Statement:** Not applicable.

**Data Availability Statement:** Data used in this study will be available upon request from the first author.

**Acknowledgments:** We would like to express our thanks to anonymous reviewers for their helpful comments on our study. We would also like to express our special thanks to Wei Bo and Gong Dianqing in the Institute of Geographic Sciences and Natural Resources Research, CAS, for their timely help.

**Conflicts of Interest:** The authors declare no conflicts of interest.

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
