# Peer review of "Qualifying Land Use and Land Cover Dynamics and Their Impacts on Ecosystem Service in Central Himalaya Transboundary Landscape Based on Google Earth Engine"

_land, doi:10.3390/land10020173_

Round 1

Reviewer 1 Report

The aim of the work is to analyze LULC changes across the whole KSL from 2000 to 2015 based on Google Earth Engine (GEE) and quantified its impacts on ESVs changes. The topic is interesting and it fill the gap on mountains areas. Furthermore, the structure and the content's sequence is well done and the only thing that i wish to suggest to the Authors consist in a better synthetic content's approach. The read of the paper require much time and appear difficult. I suggest, in my opinion, to try to summarize into the most important issues in a critical way the results and discussion paragraphs.

Author Response

Comment#1:The aim of the work is to analyze LULC changes across the whole KSL from 2000 to 2015 based on Google Earth Engine (GEE) and quantified its impacts on ESVs changes. The topic is interesting and it fill the gap on mountains areas. Furthermore, the structure and the content's sequence is well done and the only thing that i wish to suggest to the Authors consist in a better synthetic content's approach. The read of the paper require much time and appear difficult. I suggest, in my opinion, to try to summarize into the most important issues in a critical way the results and discussion paragraphs.

Response#1: We greatly appreciate your recommendations and kind comments. Mapping land cover in mountain areas are always relative difficult because of the limited satellite images and the cloudy conditions. In order to overcome the difficulties, we made some efforts based on Google Earth Engine. To better illustrate the improvement in this manuscript, we provided as more details as possible in every step and also shared our codes. This would allow replications of the experiments with different and perhaps better performing classifiers. According to your suggestions, we addressed the manuscript as below:

(1) We used professional English editing service to improve the readability.

(2) We added a short summary at the end of each paragraph in Section results and discussions.

Reviewer 2 Report

Summary:

The authors present an analysis of land use land cover change (LULC) in the Kailash Sacred Landscape (KSL) over 15 years (2000 to 2015) and it’s effect on the Ecosystem Service Values (ESV). Using Landsat imagery compositions of 3 years around the target years, 2000 and 2015, they derive the raw input features and enhance it by spectral indices, topographical features and nighttime light data. To perform the LULC on both dates they use manually selected training points that remain stable in their LULC class during the research period. A Random Forest classifier is used for classification. To assess the effect of LULC on ESVs they use elasticity.

Their claimed contributions are an exploration of the LULC dynamics between 2000 and 2015, the ESV changes based on LULC and its implication for landscape conservation and sustainable land use.

While the authors did not develop a new methodology for LULC change estimation nor its effect on ESVs they demonstrate in this mansucript a sound application that can help to translate knowledge into action and is therefore of societal significance.

General comments:

The title and abstract give a clear impression on the manuscript’s content. The choice of remote sensing data is reasonable for the given transboundary study area. While the method section provides good detail to understand how the experiments were conducted, it would have been beneficial to make the training points and source codes publicly available. This would allow replications of the experiments with different and perhaps better performing classifiers.

While the general structure of the manuscript is easy to follow, some section are difficult to read. In particular the sections 3.1, 3.2 and 3.3 are problematic as too many numeric results are mentioned directly in the text (while being listed in the tables additionally). I would suggest to abstract the gained insights and e.g. mention only the percentage values and change rates in the text to achieve a better readability.

Section 4.2 Forest fragmentation in KSL contains new results and I would suggest it to be placed under Section 3 Results.

Please use the same notation for the different LULC classes (lower case or upper case) throughout tables and text.

Please adjust formatting of mathematical symbols in the text (i.e. superscript, subscript, etc.).

The differences between the two dates in figures 2,4 and 5 are difficult to see.

Specific comments:

  1. 36: Superscript for y-1 is missing (continued throughout the manuscript).
  2. 60: The abbreviation HKH (I assume Hindu Kush Himalayan region) is not introduced.
  3. 77-81: [29, 32 and 33] seem to be very close works to this manuscript. From my understanding each of them analysed the LULC changes over different periods of time for one of the respective nations part of KLS. What are the methodological differences of [29, 32 and 33] compared this manuscript? Would one of the methods used in [29, 32 and 33] applied to the new Landsat imagery used for this manuscript obtain the same results?
  4. 101: Please clarify the term “sacred landscape”.
  5. 107: Please clarify the term “sacred lake”.
  6. 140-144: Please clarify the training point selection. I understand that the training points are manually selected by visual interpretation of the pre-processed Landsat imagery. Could you motivate the choice of manual sampling of training points over a systematic randomized approach?
  7. 142-144: The wording is a bit confusing, but I understand that the manually selected training points remain stable in their respective LULC class during the study period.
  8. 148-149: How do you deal with the varying resolutions between 30 to 100 m from Landsat-6 TM to Landsat-8 OLI?
  9. 154: The plural of index is indices.
  10. l 154: et al. ; Also what were those additional preprocessing functions? Does “ functions […] packaged together” mean nested functions?
  11. 158: et al. etc.
  12. 160: For better solve […] To solve[…]
  13. 166-169: This sentence is missing a main clause.
  14. 172: land use and land cover LULC
  15. 176-177: Adding as many features as possible is not always guaranteed to result in higher classification performance but can in fact actually degrade it with the use of (unintentionally) redundant or irrelevant features. The use of a RF classifier (l. 199) in some sense circumvents this issue, as irrelevant features are more likely to be disregarded at individual splits of a tree. If you are using the default settings of GEEs RF classifier, at an individual node of a tree a small number of features (usually the square root of the total number of features) will be considered for a split. By chance these could be NDTI, NDMI, NBR and e.g. NIR, SWIR1 and SWIR2 and thus yield highly redundant information. A feature selection step before the classifier might be reasonable to achieve better classifier performance. With a total of around 30 (?) features and a low training sample number of 80 in the least represented class it might have a positive effect. That being said, the overall accuracy of around 89% is probably good enough to support the analysis in this manuscript.
  16. 198: To summarize, please state the total number of input features for the classifier.
  17. 181: Normalized Difference Snow Moisture Index (NDMI)
  18. 197: nighttime light data
  19. 197: What nighttime light data did you use specifically and is its resolution fine enough to support your analysis? From my experience, nighttime light data has relative coarse spatial resolutions of several hundred meters.
  20. 199: about
  21. 277: It is difficult to spot the differences on this figure.
  22. 305: Is the year 1990 wrong? Should it be 2000?
  23. 320, 324, 326, 327, 329: The factor 108 is missing in the ESVs.
  24. 357: It is difficult to spot the differences on this figure.
  25. 374-375: and from 2140.04 km² to 2094.67 km², respectively.
  26. 385: Refer to the supplementary material.
  27. 403: 4.2 Forest fragamentation in KSL
  28. 403: In general to the paragraph: I am curious how fragmentation of forest relates to ESVs. Should patched or perforated forest have a lower ESV than the same area of core forest?
  29. 427: It is difficult to spot the differences on this figure.
  30. 473: What are the critiques of the benefit transfer method and why did you still choose to use it?

Author Response

Thank you for your evaluable comments.

Reviewer 3 Report

The authors present an interesting case study that is valuable from a land operations standpoint in the Kailash Sacred Landscape. Development throughout the manuscript is required before publication. Methods require use some development and clarity:

In Section 2.2 it is not clear if one of the 3 land cover systems has been followed in the paper. Is there rationale for not adopting one of the 3 systems in the region which have already been defined and published - instead of creating a highly subjective new land cover class system. Justification needs to be provided.

What kind of forest is being examined in this study? Values may vary greatly based on estimates presented by Costanza et al. Authors need to elaborate on their use of Song et al. 2017 over work of Constanza which is more widely accepted globally.

Ecological descriptions should be moved from introduction and combined in the study area. 

Results and Discussion

Section 3.3 Much of this textual information is presented in the tables. I would suggest limiting results/discussion points and leaning more on the tables as this information is redundant and difficult to follow in text format as there are so many different variables. Perhaps highlight some key variables or areas for discussion and provide some additional insight into why the RSV may be changing in those areas as well as in the overall study area.

This is an interesting study that is valuable from an operations standpoint in this region, but the methods could use some development and clarity. This is largely a case study manuscript that could benefit from more discussion on the ramifications of ESV changes (similar to what the authors have provided in the latter parts of sections 4.1 and 4.2) instead of simply stating and restating metrics in the results.

Some sections lack the succinctness and synthesis required for a manuscript publication, reading more like a graduate thesis body of work.

Other Specific Comments:  

60: HKH is undefined. Check definitions throughout

61: What are/is the conservation policies being referred to. Sentence objectives are unclear.

210: Thresholds for good and very good? Subjective.

Units are missing from the table column headings.

Figures are generally of high quality, but some text needs to be bold/increased in font size to be more readable.   

I would be happy to provide another review of the manuscript once issues in the methods section, overall synthesis/clarity, and English language style issues have been addressed.  

Author Response

Thank you for your valuable comments.
